

# CLGAN: A GAN-based video prediction model for precipitation nowcasting

Yan Ji[1,2], Bing Gong[2], Michael Langguth[2], Amirpasha Mozaffari[2], and Xiefei Zhi[1]

[1]Nanjing University of Information Science and Technology, 210044 Nanjing, China
[2]Jülich Supercomputing Centre, Forschungszentrum Jülich, 52425 Jülich, Germany

**Correspondence:** b.gong@fz-juelich.de

**Abstract.** The prediction of precipitation patterns at high spatio-temporal resolution up to two hours ahead, also known as precipitation nowcasting, is of great relevance in weather-dependant decision-making and early warning systems. In this study, we are aiming to provide an efficient and easy-to-understand model - CLGAN, to improve the nowcasting skills of heavy precipitation events with deep neural networks for video prediction. The model constitutes a Generative Adversarial Network (GAN) architecture whose generator is built upon an u-shaped encoder-decoder network (U-Net) equipped with recurrent LSTM cells to capture spatio-temporal features. A comprehensive comparison among CLGAN, and baseline models optical flow model DenseRotation as well as the advanced video prediction model PredRNN-v2 is performed. We show that CLGAN outperforms in terms of scores for dichotomous events and object-based diagnostics. The ablation study indicates that the GAN-based architecture helps to capture heavy precipitation events. The results encourage future work based on the proposed CLGAN architecture to improve the precipitation nowcasting and early-warning systems.

## 1 Introduction

Heavy precipitation can lead to numerous hazards, e.g., damages of the infrastructure, enhanced risk for human life and loss to agriculture due to flood events (Ganguly and Bras, 2003; Vasiloff et al., 2007; Li et al., 2021). Accurate predictions of strong precipitations events at high spatio-temporal resolution up to two hours ahead, also known as precipitation nowcasting, are therefore critical for establishing early-warning systems. These warning systems can in turn promote authorities in weather-dependent decision-making and enhance risk governance capabilities (Dixon and Wiener, 1993; Johnson et al., 1998; Bowler et al., 2006).

Current precipitation nowcasting mainly relies on convective-permitting numeric weather prediction (NWP) and on extrapolation techniques of precipitation patterns with the help of composite radar observations. However, NWP models still suffer from difficulties in capturing these patterns in the nowcasting time range due to model spin-up effects and due to challenges in handling non-Gaussian data in the assimilation procedure (Ravuri et al., 2021). Also, a quick model run cycle would be required. For instance, ICON-D2 is only run every 3 hours (Matsunobu et al., 2022) which makes it impossible to get quick updates in light of rapidly growing precipitation patterns that are missed in the previous forecast. Observation-based extrapola-



tion methods such as optical flow are commonly superior to NWP models for precipitation nowcasting, but also fail to capture
the underlying non-linear processes of precipitation formation, e.g. secondary triggering and aggregation (Xie et al., 2019).

Deep neural networks have gained increasing attention in the meteorological community over the last years (Reichstein
et al., 2019; Schultz et al., 2021). The growing interest can be attributed to the success stories in other domains where deep
learning (DL) proved to leverage high-level information from complex and highly non-linear data in several applications such
as autonomous driving (Hu et al., 2020), anomaly detection (Liu et al., 2018), and semantic segmentation (Garcia-Garcia et al.,
2018). Recently, video prediction models developed in the computer vision community have been explored in precipitation
nowcasting. Contemporary studies mainly applied model architectures based on u-shaped convolutional networks (U-net; e.g.
Ayzel et al., 2020; Ronneberger et al., 2015), convolutional long short-term memory cells (ConvLSTM; e.g. Shi et al., 2015),
generative (see e.g. Ravuri et al., 2021) as well as attention models (see e.g. Sønderby et al., 2020). U-Nets are thereby con-
sidered to be beneficial since they are capable to extract multi-scale features (Ronneberger et al., 2015) and since atmospheric
processes are inherently interacting on multiple spatial scales (Orlanski, 1975). To also explicitly capture temporal dependen-
cies in the underlying formation process of precipitation, recurrent ConvLSTM models are an appealing choice (Shi et al.,
2015). Thus, combining convolutional and recurrent networks with ConvLSTM-layers is advantageous in generating stable
precipitation nowcasting by learning the spatial and temporal dependency from the historical frames.

Nevertheless, these models have problems with handling the statistical nature of precipitation, especially when a pixel-
wise loss function is applied for the optimization process during training (Shi et al., 2017; Ayzel et al., 2020). Although log
transformation, importance sampling and weighting towards heavier precipitation targets are appropriate to govern the right-
skewed Gamma distribution of precipitation rates (e.g. Ravuri et al., 2021), the inherent uncertainty in quasi-chaotic processes
at meso-scale typically leads to unrealistically smooth precipitation patterns in the forecasts. While this issue is well known
in many other video prediction tasks (Mathieu et al., 2015; Ebert et al., 2017), it is of particular relevance in precipitation
nowcasting. The high spatio-temporal variability seen in the observational (real) data cannot be maintained and thus, heavy
precipitation events are barely captured by models applying a pixel-wise loss. Generative models which train a generator and
a discriminator adversarially (GAN models) are considered to be a potential solution for such applications (Goodfellow et al.,
2020). By forcing the generator to fool the discriminator which aims to distinguish between real and generated data, these
models succeed in maintaining the statistical properties of the underlying data (Oprea et al., 2020). Likewise, GAN models
can be easily extended to provide ensemble forecasts which allow a quantification of the forecast uncertainty (Mordido et al.,
2018).

Although great progress has been achieved in a series of recent work (e.g. Ravuri et al., 2021; Gong et al., 2022), there is
controversy regarding how to understand the contribution of each component within the sophisticated models. A meticulous
analysis of the precipitation nowcasting in terms of the attributions of each precipitation object is further required. Motivated by
this, we build a simple but efficient and easy-to-understand video prediction model, CLGAN (Convolutional Long short-term
memory Generative Adversarial Network; see Fig. 1), for the nowcasting task and multiple application-specific verification
methods including object-based approach are used to evaluate the model performance. CLGAN is proposed to leverage the
advantages of different DL model architectures. The generator combines the U-Net with a ConvLSTM cell to abstract spatial





features on multiple scales, while the temporal dependency of precipitation patterns is also preserved. The generator network

is then trained adversarially to attain precipitation forecasts resembling observed data. For our nowcasting application, we deploy a gridded dataset with a temporal resolution of 10 minutes aggregated from automatic weather station (AWS) gauges over Guizhou, China. The predictive performance of the proposed model architecture is then accessed in a comprehensive evaluation based on metrics designed for precipitation nowcasting as mentioned. The evaluation also involves a comparison against a simplistic persistence forecast, the conventional optical flow model DenseRotation (Ayzel et al., 2019) as well as

two baseline video prediction models, a standard ConvLSTM (Shi et al., 2015) network and a up-to-date competing model PredRNN-v2 (Wang et al., 2021).

With this, the main contributions of our study are:

- – An efficient and easy-to-understand architecture CLGAN leveraging the merits of U-Net, ConvLSTM and GAN models is proposed to generate perceptually realistic precipitation forecasts.

- – A new 10-minute-level precipitation dataset based on AWS gauges (Guizhou AWS_ML precipitation dataset) is built for machine learning experiments.

- – Nowcasting of heavy precipitation events is improved with a comprehensive verification.

- – A sensitivity analysis is performed on the adversarial loss sheds light on the interaction between the generator and the discriminator.

## 2   Related Work and baseline models

**Conventional Methods.** The simplest approach to generate a precipitation 'forecast' is to deploy the Eulerian persistence. For this, the most recent available observation, usually a radar composite, is used and then replicated several times for the future steps. This approach is quite accurate for very short lead times, but obviously fails to provide meaningful forecasts in a quickly evolving system such as the atmosphere for time scales beyond several minutes. Thus, the related forecast quality can

be considered as the minimum level for a prediction model to be useful.

Conventional precipitation nowcasting systems typically use a Lagrangian framework to predict the development of precipitation patterns. Although this framework often assumes persistence of the precipitation features' intensity and displacement, it is still capable to outperform mesoscale NWP models in precipitation nowcasting (Sun et al., 2014). The Lagrangian method applies a two-step approach where the precipitation features are first tracked and then extrapolated to future time steps (Austin

and Bellon, 1974). Typically, the tracking step is accomplished with the help of optical flow methods that infer the motion of patterns from consecutive images. For precipitation nowcasting, radar composite images are subject to a tracking algorithm such as cross-correlation tracking (Rinehart and Garvey, 1978; Grecu and Krajewski, 2000; Zahraei et al., 2012) or centroid tracking techniques (Zahraei et al., 2013). The tracked objects are then applied to different extrapolation schemes, e.g. image warping (Wolberg, 1990), constant-vector advection (Bowler et al., 2004) or semi-Lagrangian scheme (Germann and Zawadzki,



2002). With this two-step approach, several operational precipitation nowcasting systems have been established over the globe in the last three decades, such as the Thunderstorm Identification Tracking, Nowcasting (TITAN; Dixon and Wiener, 1993), the Storm Cell Identification and Tracking (SCIT; Johnson et al., 1998), and the Short-Term Ensemble Prediction System (STEPS; Bowler et al., 2006) (see Wilson et al., 2010, for a review on operational systems).

Recently, Ayzel et al. (2019) implemented a set of advanced optical flow models into an open-source Python library called
*rainymotion*. Two different groups of methods are part of this library from which we show more interest in the DenseRotation model that performs best in their study. The tracking algorithm of this model is based on the Dense Inverse Search algorithm proposed in Kroeger et al. (2016) which provides an estimate of the motion of each pixel based on two consecutive radar images. The extrapolation is then performed with a semi-Lagrangian advection scheme (Germann and Zawadzki, 2002), which is capable to represent rotational motions.

In our paper, the simplistic Eulerian persistence model and the DenseRotation model serve as classical baselines. The two conventional models are applied here to see how much benefits can be obtained by performing DL-based video prediction methods for the precipitation nowcasting task.

**Video Prediction Method.** As already mentioned, the application of deep learning techniques in the meteorological community has gained momentum over the recent years. In particular, several studies have started to explore these techniques to tackle
the precipitation nowcasting problem. By formulating precipitation nowcasting as a sequence prediction task, Shi et al. (2015) proposed a network of ConvLSTM cells which apply a convolution in the recurrent layers of the vanilla LSTM to capture spatio-temporal features in the underlying data. Their 2-layer ConvLSTM network was able to outperform Real-time Optical flow by Variational methods for Echoes of Radar (ROVER; Woo and Wong, 2017), an operational precipitation nowcasting system based on optical flow methods with a semi-Lagrangian advection scheme. Shi et al. (2017) further extended the recurrent
cells of GRUs with non-local neural connections and proposed the Trajectory GRU (TrajGRU) model which enables learning of location-variant structures of precipitation.

Besides, Wang et al. (2017) advanced the application of ConvLSTM networks and proposed the predictive recurrent neural network (PredRNN). They deployed a stack of recurrent layers that feature a zigzag memory flow and involve an explicit spatio-temporal memory state. In this way, they enable an explicit communication of abstracted spatio-temporal features between
different levels of the recurrent network which yields improved precipitation predictions. While this approach already provided promising results, the PredRNN model was further developed to PredRNN-V2 (Wang et al., 2021). The advances comprise the implementation of a decoupling loss ST-LSTM to enhance featuring of the spatio-temporal variations and a new, improved long-term modeling strategy. The model attains remarkable improvements when applied to multiple datasets including the radar echoes.

Meanwhile, other network architectures were also explored in scope of precipitation nowcasting. Several studies started to apply fully convolutional U-Net architectures which are capable to abstract and exploit spatial features on different scales via a hierarchical encoder-decoder network with skip connections between its branches. Notably, the RainNet-architecture proposed by (Ayzel et al., 2020) proved to significantly outperform optical flow based nowcasting methods for weak precipitation events. However, their network tends to provide smooth precipitation fields and therefore fails to provide added value for more intense





precipitation events with a rain rate above $10\,\mathrm{mm/h}$. Recently, a deep generative model for the probabilistic precipitation nowcasting was proposed and showed state-of-the-art performance for the task (Ravuri et al., 2021).

All these studies demonstrate that deep neural networks have the potential to provide added value for precipitation nowcasting. In our study, we mainly focus on further improving the predictions of strong precipitation events and therefore choose a simple ConvLSTM (Shi et al., 2015) and the advanced PredRNN-v2 (Wang et al., 2021) model for competing with our newly
proposed model architecture.

## 3    Method and Data

### 3.1    Our model CLGAN

In the following, we present in more detail our proposed CLGAN architecture. Since CLGAN aims to benefit from ConvLSTM-models, the U-Net architecture and the GAN models, we first introduce its components separately to provide a deeper under-
standing and reasoning for the chosen model architecture.

#### 3.1.1    ConvLSTM

The basic formulas of the ConvLSTM cell which describe the gated update procedure for the hidden and the cell state are provided in Shi et al. (2015) and are therefore not repeated here. The objective function of a ConvLSTM model typically constitutes the classical $\mathcal{L}^2$ reconstruction loss. This loss measures the distance between the predicted and the target (ground
truth) data on grid point (or pixel-wise) level and can be written as

$$\mathcal{L}^2(G) = \left\| \overrightarrow{X}_{T_0+1:T} - \hat{\overrightarrow{X}}_{T_0+1:T} \right\|_2 \tag{1}$$

where $\overrightarrow{X}$ and $\hat{\overrightarrow{X}}$ are two-dimensional tensors for the ground truth and the predicted data, respectively. $T_0$ represents the end of the input sequence, so that the model is optimized on the loss over the prediction sequence which ends at time step $T$. These tensors comprise $N \times M$ grid points in zonal and meridional direction of the domain of interest. Note that the loss only
operates on the predicted data sequence which starts with the first time step after the end of the input data sequence $T_0+1$. The complete data sequence, i.e. input and prediction, involves $T$ time steps.

#### 3.1.2    U-Net

The U-Net model was originally applied for biomedical image segmentation (Ronneberger et al., 2015) and is therefore designed as a powerful feature extractor on various spatial scales. As illustrated in Fig. 1a, it can be decomposed into a compress-
ing and an expansive path that are bridged by skip connections. The contracting path can be seen as an encoder which converts the high-resolved data into coarse-grained features using convolutional and pooling layers. The expansive path, acting as a decoder, applies deconvolutional layers to convert back to the original spatial resolution, of which the number of data points are $N \times M$. Usually, several pooling and deconvolution layers are applied to allow feature extraction on different spatial scales.





To avoid the vanishing gradients issue and to allow a direct information flow of specific spatial features, skip connections are implemented at every scale-specific feature extraction level (Drozdzal et al., 2016).

In a video prediction application, the data at time step $T$ enters the encoder to produce a forecast at time step $T+1$ with the decoder. By doing so, no long-term information is explicitly conveyed as with the ConvLSTM-model. Since heavy precipitation events are rare, but of high relevance for nowcasting, different techniques are usually applied to encourage deep neural networks in predicting events on the right tail of the underlying PDF. Log-transformation converts the right-skewed Gamma distribution of precipitation data (e.g. RainNet in Ayzel et al., 2020) into a Gaussian-like distribution which puts strong precipitation events closer to the center of mass in probability space. Stronger weighting on higher precipitation rates and importance sampling can further support the optimization efficiency with respect to heavy precipitation events (Ravuri et al., 2021). Nonetheless, U-Nets and ConvLSTM modes still tend to produce too smooth precipitation patterns, thereby failing to capture the relevant strong precipitation events.

### 3.1.3 Generative adversarial networks

To enforce a closer agreement of the generated data with the ground truth, GAN models were proposed by Goodfellow et al. (2020). A GAN model consists of a generative network $G$ (generator) and a discriminative network $D$ (discriminator) which aims to assign a probability of 1 to real data and a probability of 0 to generated data. While the discriminator is optimized to distinguish between both kinds of inputted data, the generator is encouraged to fool the discriminator. Thus, the GAN applies the binary cross-entropy loss as the objective function which enters a minimax-game:

$$G^{\star} = \arg\min_{G}\max_{D}\mathcal{L}^{GAN}(G,D)$$

$$\text{with } \mathcal{L}^{GAN}(G,D) = \mathbb{E}_{\vec{X}_{1:T}}\left[\log D(\vec{X}_{T_0+1:T})\right] + \mathbb{E}_{\vec{X}_{1:T}}\left[\log(1-D(G(\vec{X}_{1:T_0})))\right]. \tag{2}$$

Here, the generator is conditioned on the input data sequence $\vec{X}_{1:T_0}$. Since generator and discriminator are trained adversarially, the generator is encouraged to create predictions that share the same statistical properties as the ground truth data. This is considered to be useful for generating realistic precipitation forecasts which should exhibit the high spatial variability of the observed data (Ravuri et al., 2021; Price and Rasp, 2022; Harris et al., 2022).

### 3.1.4 Convolutional LSTM GAN (CLGAN)

To combine the merits of a GAN model with the strong spatio-temporal feature extraction capacities of U-Nets and ConvLSTM models, we set up the generator $G$ as follows: The generator constitues a three-level U-net following Sha et al. (2020). Each level of the encoder comprises two convolutional layers followed by max-pooling with a 2x2-kernel to reduce the spatial dimensionality in the next layer. The number of channels is thereby increased by a factor of two in each level. A ConvLSTM cell with 64 filters is deployed to implement recurrency at the bridge between the encoder and decoder. The decoder then reverts the encoded data to the input resolution with the help of deconvolutional layers. Furthermore, skip connections among the encoder and decoder are added at each level of the U-net. The discriminator $D$ consists of 3D fully convolutional layers with batch



normalization which allow us to encode both, the temporal and spatial dimension of the data sequence. Again, max-pooling is used to compress the data which finally gets concatenated to fully connected layers (see Fig. 1b). The forecast sequence of $G$ $\hat{\vec{X}}_{T_0+1:T}$ as well as the corresponding ground-truth sequence $\vec{X}_{T_0+1:T}$ are taken as the inputs for the discriminator $D$.

In this study, the generator is trained by combining the adversarial loss $\mathcal{L}^{GAN}$ with the reconstruction $\mathcal{L}^2$-loss:

$$G^\star = (1-\lambda)\mathcal{L}^{GAN}(G,D) + \lambda\mathcal{L}^2(G) \quad \text{with } \lambda \in [0,1]. \tag{3}$$

This ensures that the prediction remain close to the ground truth. The relative weight of the reconstruction loss $\lambda$ is set to $0.99$ which proves to balance the contributions from both loss components in the following experiments. Training of the model is performed with the Adam optimizer (Kingma and Ba, 2014) over eight epochs with a batch size of 32.

## 3.2 Guizhou AWS_ML precipitation dataset

In addition of the widely used remote sensing data, e.g. radar composite images, measurements from densely-distributed auto-
matic weather stations can serve as an alternative in the data-driven weather forecasting. In this study, minute-level precipitation measurements by rain gauges of AWS over Guizhou, China (Guizhou AWS_ML precipitation dataset) is collected for the precipitation nowcasting task. Guizhou is a mountainous and rainy province located in southwest China (see Fig. 2a) where mudslides happen frequently during summer-times. For instance, the region was affected by a severe rainstorm in September 2020, when some regions experienced more than $1500\,\text{mm}$ rainfall within 20 days. Accurate precipitation nowcasting, especially for
heavy precipitation, is crucial to reduce damages from these events. Hence, the Guizhou AWS_ML precipitation dataset is established for better simulation of precipitation with data-driven approaches. The AWS locations comprise 93 national basic stations and 1740 automatic weather stations (see Fig. 2b). Among other meteorological quantities (2-m temperature, 10-m wind, surface pressure and relative humidity), the AWS measures precipitation at a high observation frequency (every minute) and the data is provided between 1st Jan 2015 and 31st Dec 2019 by Guizhou Meteorological Bureau.
Several preprocessing steps are conducted for preparing the dataset of our experiment. First, the precipitation data is accumulated over 10 minutes which still constitutes a reasonably high temporal resolution. To obtain a gridded dataset, the observations are then interpolated bilinearly onto a regular, spherical grid. The target grid comprises $N \times M = 48 \times 40$ data points in zonal and meridional direction, respectively, and covers a domain from $103.625°\text{E}$ to $109.5°\text{E}$ and $24.625°\text{N}$ to $29.5°\text{N}$ at $0.125°$ resolution. To obtain the data needed for training our CLGAN and the baseline models (see Sec. 2 and
3.1.4), we generate sliding sequences of 24 consecutive gridded data samples (frames) which comprise a temporal period of 240 minutes. 120 minutes (12 frames) of each sequence serve as input to predict the next 120 minutes (12 frames). Since there are many periods with no or only weak precipitation, we furthermore only select sequences whose averaged precipitation rate exceeds the empirical $60\%$-quantile of the complete dataset. This results in 35054 sequences for the subsampled dataset. Finally, a log-transformation is applied on each sequence to make the data more Gaussian-like. The log-transformation reads
as follows: $x' = \ln(x+\varepsilon) - \ln(\varepsilon)$, where $\varepsilon$ is a small constant (here 0.01). We use the data from 2015 to 2017 for training, the data of 2018 for validating and the data of 2019 for testing.





### 3.3 Verification methods

As pointed out in Schultz et al. (2021) and more specifically for precipitation in Leinonen et al. (2020), precipitation nowcasting should be evaluated in terms of application-specific scores. This is due to the unique statistical properties of precipitation rates,
but also due to the chaotic atmospheric processes which underpin the formation of precipitation. Additionally, we would like to emphasize that a single score alone can barely evaluate the model performance applied to high-dimensional data (Wilks, 2011). Therefore, we take several evaluation metrics into account to provide a comprehensive overview.

The first considered family of evaluation metrics is well established for continuous quantities in the meteorological community. The root mean square error (RMSE) measures the distance between the predicted and the observed field on a grid-point
level. The correlation coefficient (CC) measures the association or the linear relationship between the two fields. A perfect correlation would result in $CC = 1$, while $CC = 0$ indicates no linear relationship between forecast and observation on grid-point level.

The second set of scores is built on dichotomous events which are obtained by thresholding the gridded precipitation fields. A $2 \times 2$ contingency table is commonly used to show the frequency of "yes" and "no" forecasts and occurrences and give a joint
distribution for events with a precipitation rate exceeding a given threshold $t_{pr}$. According to the elements in the contingency table, a variety of categorical statistics can be computed to evaluate the dichotomous forecasts in particular aspects. Critical Success Index (CSI), also known as Threat Score, measures the fraction of hits with respect to the number of occurrences where the events are either forecasted or observed. The frequently applied Equitable Threat Score (ETS) is a variant of the CSI and explicitly accounts for random forecasts which perform well just by chance (Wilks, 2011).
However, due to the highly non-linear and complex processes causing precipitation formation, scores acting on grid-point level are prone to penalize predictions which recover the high spatial variability, but fail to match exactly the observed precipitation field. The issue leads to the double penalty problem where the model gets penalized twice, once for missing the exact placement of a precipitation event and once for shifting it spatially (Ebert, 2008). To relax the requirement for exact spatial matching, the Fractions Skill Score (FSS) is computed here as a fuzzy verification metric (Roberts, 2008). Similar with the CSI
and ETS, the FSS operates on dichotomous events, but allows for spatial shifts by considering a local neighborhood around each grid point. Within this neighborhood, the fractional coverage of the precipitation events is calculated for both, the predictions and the observations. Let $\langle f(m_0) \rangle_s$ ($\langle f(o_0) \rangle_s$) denote the fraction of event grid boxes within the local neighborhood of size $s$ around the grid point $i$ in the model forecast (observation). The Fractions Brier Score (FBS) is given by:

$$FBS = \frac{1}{N} \sum_{i=1}^{N} \left[ ((\langle f(m_0) \rangle_s)_i - (\langle f(o_0) \rangle_s)_i \right]^2. \tag{4}$$

which quantifies the quadratic difference between the prediction and the observation for all $N$ grid points of the domain. The final FSS is then obtained with:

$$FSS = 1 - FBS/FBS_{worst}. \tag{5}$$





where $FBS_{worst}$ is the sum of the squared fractions of events in the prediction and in the observation. Higher FSS-values indicate better forecast, while it can be shown that a forecast becomes 'useful' when $FSS \geq 0.5$ is attained for a given neigh-

borhood scale $s$ (typically expressed in terms of squares with an edge length of $n$ grid points).

Nonetheless, FSS also does not capture the spatial precipitation patterns since each grid point in the neighborhood is treated equally and no check for spatial coherence is undertaken. Thus, we additionally perform an object-based diagnostic evaluation, called MODE (Johnson and Wang, 2012; Johnson et al., 2013; Ji et al., 2020), to focus on pattern attributes such as location, area and shape. To obtain the desired attributes, a convolutional filter of size $k$ is first applied over the precipitation field.

Afterwards, objects are defined by applying an threshold on the precipitation rate $t_{Pr}$ and on the object area $t_A$. A fuzzy logic scheme is then used to merge and pair precipitation objects in the predicted and observed precipitation field. Finally, the object-based threat score (OTS; Johnson and Wang, 2012) is computed to verify how well the predicted precipitation patterns match the observed ones. Here, we choose the object area, the centroid location and the object shape (aspect ratio and orientation angle) as target attributes for computing the OTS.

All the mentioned verification methods are listed in Table 1 with a brief description. The details can be found in the corresponding references.

To ease the comparison between the baseline models and the simplistic persistence forecast, we furthermore calculate skill scores (except for the FSS). In general, a skill score $SS$ can be constructed by considering the target score $S_m$ of the model, the score obtained with the reference forecast $S_{ref}$ and the perfect score $S_{perf}$:

$$SS = \frac{S_m - S_{ref}}{S_{perf} - S_{ref}}. \tag{6}$$


The higher $SS$ is, the better the model performs against the reference score. Perfect models thereby obtain $SS = 1$, while inferior models show up with $-\infty < SS < 0$. Note, that $S_{perf} = 0$ holds for the RMSE, whereas the other scores under consideration attain $S_{perf} = 1$. Since the size of our dataset is not unlimited, we also apply a block bootstrapping procedure to estimate sampling uncertainty (Efron and Tibshirani, 1994). The block bootstrapping procedure accounts for autocorrelation

between the sliding sequences and thus, divides the dataset into non-overlapping blocks before resampling of the blocks with replacement is performed. Here, we set the block length to 10 hours (60 frames) and perform 1000 block bootstrapping steps.

## 4 Results

The performance of precipitation nowcasting using DenseRotation and persistence as well as video prediction models is discussed in the following section.

In Fig. 3a∼d, our model is compared to baseline models in terms of skill scores for the grid-point level evaluation metrics (CC, RMSE, CSI, and ETS). The skill scores are calculated by defining the Eulerian persistence as reference forecast. It is seen that the deep learning models (i.e. ConvLSTM, PredRNN-v2 and CLGAN) as well as the optical flow model DenseRotation outperform the persistence forecast after 20 minutes lead time in terms of the continuous scores (CC and RMSE). Among the video prediction models, PredRNN-v2 is superior over the others over the first 20 minutes, while ConvLSTM performs best for





**Table 1.** Summary of the verification methods used in the paper

| Verification method | Description | Formula or Reference | Notes |
|---|---|---|---|
| Root Mean Square Error | the average magnitude of the forecast errors | $\sqrt{\frac{1}{N}\sum_{i=1}^{N}(Y_i - Y_i')^2}$ | $[0, +\infty)$; |
| Correlation Coefficient | the correspondence between the forecast and observed values | $\frac{\sum_{i=1}^{N}(Y_i - \bar{Y}_i)(Y_i' - \bar{Y}_i')}{\sqrt{\sum_{i=1}^{N}(Y_i - \bar{Y}_i)^2}\sqrt{\sum_{i=1}^{N}(Y_i' - \bar{Y}_i')^2}}$ | $[-1, 1]$; |
| Critical Success Index | the correspondence between the forecast "yes" events and observed "yes" events | $\frac{hits}{hits + misses + false\ alarms}$ | $[0, 1]$, 0 means no skills; |
| Equitable Threat Score | the correspondence between the forecast "yes" events and observed "yes" events (accounting for hits due to chance) | $\frac{hits - hits_{random}}{hits + false\ alarms}$ $hits_{random} =$ $\frac{(hits + misses)(hits + false\ alarms)}{total}$ | $[-1/3, 1]$, 0 means no skills; |
| Fractions Skill Score | the spatial scales at which the forecast resembles the observations | (Roberts and Lean, 2008) | $[0,1]$, the smallest window size for which FSS$\geq$0.5 can be considered as "skillful scale"; |
| Object-based Threat Score | the similarity between the forecast objects with the observed ones according to a series of attributes | (Davis et al., 2006) | $[0, 1]$, 0 means complete mismatch and 1 means perfect match; |

the longer lead times. CLGAN is not so competitive for RMSE and CC as PredRNN-v2 and ConvLSTM, while still outperforms the traditional optical flow model DenseRotation. However, when comparing the performance for the dichotomous scores (CSI and ETS) focusing on heavy precipitation events (here the threshold is 8mm/h), CLGAN is superior to the other models across all the lead times (see Fig. 3c∼d). The optical flow model DenseRotation performs well in the first 40 lead minutes while its skill scores decrease rapidly. By contrast, the advanced deep learning model PredRNN-v2 shows more potential for the longer

lead times. Especially, although ConvLSTM outperforms on the continuous scores, it can barely capture the heavy precipitation events and is even inferior to the persistence forecast for the first lead hour. The comparisons of the model performance show that CLGAN is superior in terms of scores for dichotomous forecasts (CSI and ETS) while is less competitive in terms of RMSE.

To further investigate the model performances, we now turn our attention to the spatial verification scores, the FSS and

the MODE framework. Fig. 4a shows the model performance on FSS at 60 lead minutes by the persistence method (used as the reference forecast). The FSS is computed based on different neighbourhood sizes and thresholds of hourly precipitation rates. Specifically, the neighbourhood scale $s$ in kilometers varies along the $x$-axis. The FSS-values for $s$ attaining values of



approximately 41, 69, 96, 152, and 290 kilometers (square boxes of 3, 5, 7, 11 and 21 grid points), respectively, are plotted and marked as box-whisker plots for varying precipitation thresholds. As the threshold increases, FSS decreases, indicating

that the persistence forecasts become increasingly imprecise for stronger precipitation events with a given spatial scale. For precipitation events exceeding $t_{Pr} = 8$ mm/h, the persistence forecasts is considered useful ($FSS \geq 0.5$; see e.g. Roberts, 2008) for a neighborhood scale of $s \approx 69$ km. Thus, the spatial accuracy of capturing these events is already fairly degraded and the neighborhood scale of $s \approx 69$ km (5 grid points) is applied to compute the corresponding FSS. With persistence as the reference forecast, Fig. 4b compares the models' performance in terms of FSS for heavy precipitation forecasting. It is seen

that all baseline models except ConvLSTM can remarkably improve the spatial forecasting of such events, especially for the longer lead times. Among them, CLGAN is superior to the others at all the lead times. DenseRotation performs well in the first lead hour while PredRNN-v2 is promising for the further lead times.

To fully access the performance of the models in predicting precipitation spatial attributes, i.e. area, location and shape, the MODE verification framework is applied here. In the following, we present conditional quantile plots for the object area, for

the location of the object centroid in east-west and north-south direction, for the aspect ratio and for the orientation angle of the precipitation objects to show more details of predicted and observed precipitation objects. These plots visualize the joint distribution of the predictions and forecasts in a compact manner by applying a factorization into a conditional and marginal distribution (Murphy and Winkler, 1987; Wilks, 2011). Fig 5 illustrates the joint distribution in terms of the likelihood-base rate factorization. While the solid lines illustrate the forecasts conditioned on the observations for all models, the marginal

distribution of the observations is plotted as a histogram. It can be seen that CLGAN and PredRNN-v2 are able to capture object area fairly well. Only objects consisting of 90 to 130 grid points ($\sim 16,000 km^2$) are slightly underestimated (see Fig. 5a). However, the other competitor models perform remarkably worse. The location of object centroids (Fig. 5b∼c) is generally well captured by all models. Stronger deviations are visible near the lateral boundaries, especially in east-west direction. The aspect ratio and the orientation angle are used to assess the predicted precipitation shape in Fig. 5d∼e. CLGAN

shows slight improvements over the other models where the central parts of the orientation angle and aspect ratio are well calibrated. However, larger deviations from the 1:1 reference line are obtained near the tails of the conditional distributions. It indicates that further studies on simulation of precipitation shape are required.

To gain further insight into the realism of precipitation nowcasting, a heavy precipitation event occurred on June 12th 2019 is visualized as an example (see Fig. 6) to compare the model performance with an 'eyeball' analysis. Fig. 6a shows the observed

precipitation rates in mm/(10 min) for every 20 minute over the forecast period starting at 06:50 BJT. It is seen that a fairly strong precipitation system moves from west to east while it further intensifies. The predictions of the different models are presented as difference plots in Fig. 6b∼f. For the first 60 minutes, persistence and DenseRotation show up with the smallest discrepancies. However, for longer lead times, clear dipole structures in the difference plots indicate that the movement of the system is not detected. It demonstrates that the persistent Lagrangian framework is not sufficient for the long lead times and

more advanced models are required to capture the long-term dependence.

While the deep learning models also show up with increasing differences with lead times, they show better capability of capturing the movement and the intensification of the precipitation system (see Fig. 6d∼f). PredRNN-v2 trends to over-estimate



the precipitation intensity which causes large coherent areas of positive differences. ConvLSTM and CLGAN perform better
with smaller discrepancies. Compared to ConvLSTM, the difference plot of CLGAN contains more fine 'cells' which indicates
that CLGAN can generate more details of the precipitation system.

As shown in Eq. 3, the loss function used in our CLGAN model consists of two terms: the adversarial loss $\mathcal{L}^{GAN}$ and the
reconstruction loss $\mathcal{L}^2$. To assess the contribution of each loss term on the forecasts performance, sensitivity experiments on
the weight of the reconstruction loss in CLGAN were carried out. Larger weight of the adversarial loss (smaller weight of
reconstruction loss) is equivalent to a stronger contribution made by the GAN component. Fig. 7 presents the results of the
CLGAN model with different weights assigned to the $\mathcal{L}^2$ loss. It is seen that RMSE is increased when reducing the weight
of $\mathcal{L}^2$ loss (Fig. 7a). However, the scores for dichotomous events and fuzzy verification framework reveal improvements. The
model using a pure reconstruction loss ($\lambda = 1$ in Eq 3) performs significantly worse than the model applying an adversarial loss
in terms of CSI and FSS (Fig. 7b∼c). Similar results are obtained in terms of the OTS (see Fig. 7d). The results of sensitivity
experiments indicate that the adversarial training with the GAN-component encourages the model to generate forecasts which
are more similar to the ground truth data. Although a slight increase in RMSE is seen, a relatively stronger contribution of
the GAN-component helps to capture the statistical properties of the observed precipitation (on the tail as well as their spatial
attributes).

## 5  Conclusion and Discussion

Video prediction methods have shown good skills in precipitation nowcasting. Inspired by the success, a novel architecture
CLGAN is proposed in this work, which leverages the merits of U-Net, ConvLSTM and GAN components, to generate high
quality precipitation predictions up to 2 hours over Guizhou, China. The Eulerian persistence is used as the baseline and a
conventional optical flow model DenseRotation, as well as two video prediction models (ConvLSTM and PredRNN-v2) are
performed as competitor models. A Guizhou AWS_ML precipitation dataset is newly built for the task, on minute-level pre-
cipitation measurements from the AWS gauges. The model performance is comprehensively evaluated by a series of domain-
specific evaluation metrics, including the point-by-point and object-based verification methods. The results demonstrate that
DL-based video prediction models are generally superior to the conventional methods, especially for the lead times over 60
minutes. However, the use of losses acting on grid-point level (e.g. $\mathcal{L}^1$ or $\mathcal{L}^2$ loss) diminishes the capability of models in cap-
turing heavy precipitation events. Since heavy precipitation events are strongly under-represented in the data during training,
the models using grid-point level loss favor to predict weak precipitation rates, which avoids to yield large loss contributions.
By contrast, the GAN-component of CLGAN encourages the generator to create predictions that share the statistical properties
of observed precipitation, which makes it superior to the baseline and competitor models in dichotomous and spatial scores for
heavy precipitation events.

Our results indicate that video prediction models with deep neural networks have more potential to capture the temporal-
spatial information embedding in the high-dimension weather data. By learning the statistical dependency within the continuous
sequence of precipitation data, video prediction models can well simulate the precipitation patterns up to 2 hours. Since the

NWP models suffer from the spin-up issue in the first 6 hours and the conventional approaches are too weak to capture the long-term dependency, video prediction models show potential as a promising and reliable way for precipitation nowcasting. Nevertheless, it is still challenging to capture the shape features of precipitation by DL-based models in terms of the MODE scores. It will be very interesting to integrate the domain-specific evaluation metrics for spatial forecasts (e.g. FSS and MODE) as the loss function in DL-based models for precipitation nowcasting. Additionally, we also see that a trade-off between evaluations on grid-point level and object-based evaluations exists when the adversarial loss is varied. Grid search for the optimal combination of loss function coefficients is required to generate realistic forecasts with a low margin of error.

Apart from precipitation, 2-m temperature, 10-m wind, surface pressure and relative humidity can also be obtained from AWS gauges and used in our follow-up study. These variables can be further embedded as the additional predictors to exploit explicitly physical knowledge. It shows that the corresponding predictors and physical constraints are proven beneficial and can greatly improve the simulation of the targeted variable (Daw et al., 2017; Gong et al., 2022). In addition, as a underlying application, our CLGAN can further provide probabilistic forecasting by adding random noises. By transferring the deterministic forecasting to the probabilistic forecasting, the model has more chance to capture the heavy or even extreme precipitation events and better quantifies the uncertainty of prediction (see e.g. Ravuri et al., 2021).

*Code and data availability.* The Guizhou AWS_ML precipitation dataset and the exact version of the video prediction models used in this paper are archived on Zenodo. The dataset and scripts can help users to reproduce the results on their local machines or high-performance computers. By using these data and models, it is highly recommended to follow the README.md file of the code repository to run the end-to-end workflow.

*Author contributions.* Yan Ji, Bing Gong and Michael Langguth equally contributed to this work. Yan Ji and Bing Gong contributed to the method development and perform the experiments. Yan Ji wrote the manuscript draft and all authors reviewed and edited the manuscript in several iterations.

*Competing interests.* The authors declare that they have no conflict of interest

*Acknowledgements.* The authors acknowledge funding from the DeepRain project under grant agreement 01 IS18047A from the Bundesministerium für Bildung und Forschung (BMBF), from the European Union H2020 MAELSTROM project (grant No. 955513, co-funding by BMBF), and from the ERC Advanced grant IntelliAQ (grant no. 787576). We thank Dexuan Kong for preparing the datasets used in our research, as well as Martin G. Schultz for the helpful scientific discussions.





(a)

**Generator**

(b)

**Discriminator**

(c)

**CLGAN**

**Figure 1.** The details of the proposed CLGAN model; (a) Generator; (b) Discriminator; (c) CLGAN: the inputs are image sequences from $t-m+1$ to $t$, $c$ is the number of channels of inputs, $ngf$ is the number of filters in the first layer of U-Net, the outputs $Y'$ are the predicted sequences from $t+1$ to $t+n$ and $Y$ is the corresponding ground-truth; $\mathcal{L}^2$, $\mathcal{L}^G$ and $\mathcal{L}^D$ are the reconstruction loss, generator loss and discriminator loss respectively.



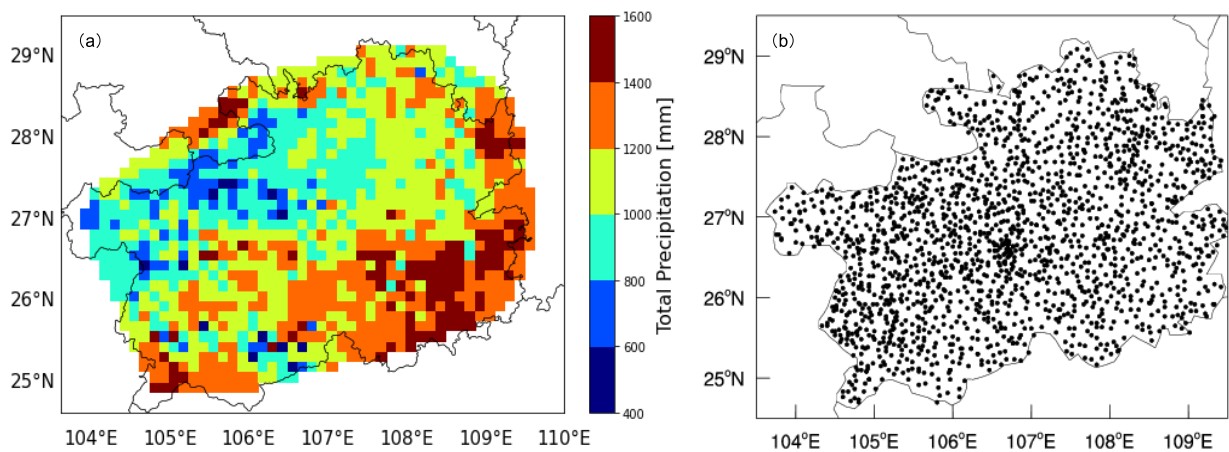

**Figure 2.** (a) Annual average cumulative precipitation in Guizhou from 2015 to 2019; (b) The spatial distribution of AWS over Guizhou.



**Figure 3.** Skill scores of (a) CC, (b) RMSE, (c) CSI and (d) ETS averaged over the testing period with the Eulerian persistence as the reference forecast. The threshold $t_{pr}$ of CSI and ETS is 8mm/h.



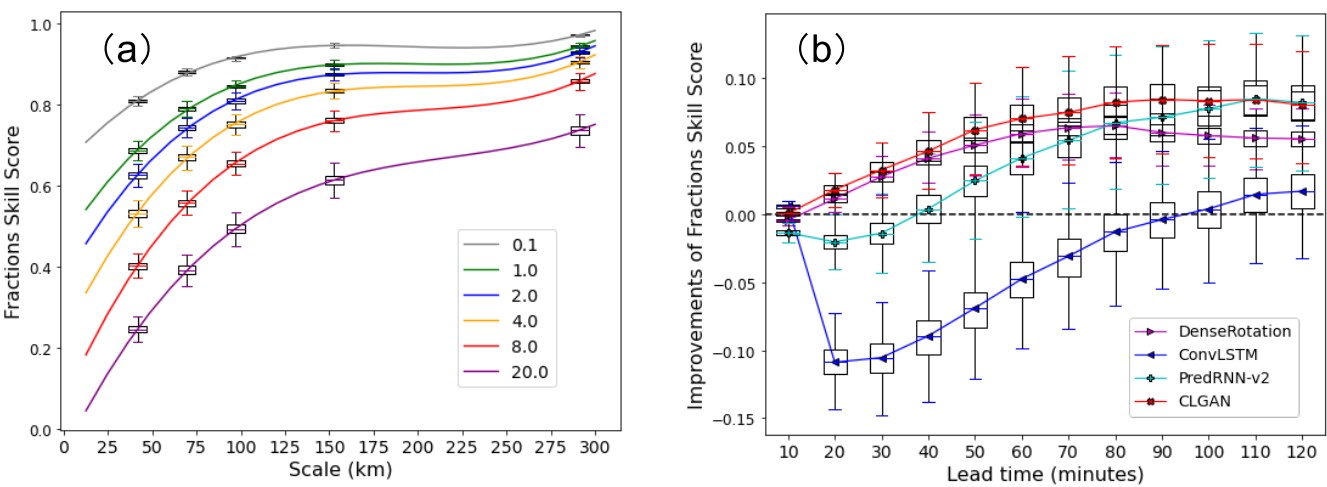

**Figure 4.** (a) FSS of persistence for different scales and intensity thresholds at 60 lead minutes for precipitation nowcasting. (b) Improvements of different models compared with persistence in terms of FSS with a threshold $t_{pr}$ of 8mm/h and a neighbourhood size of 5.

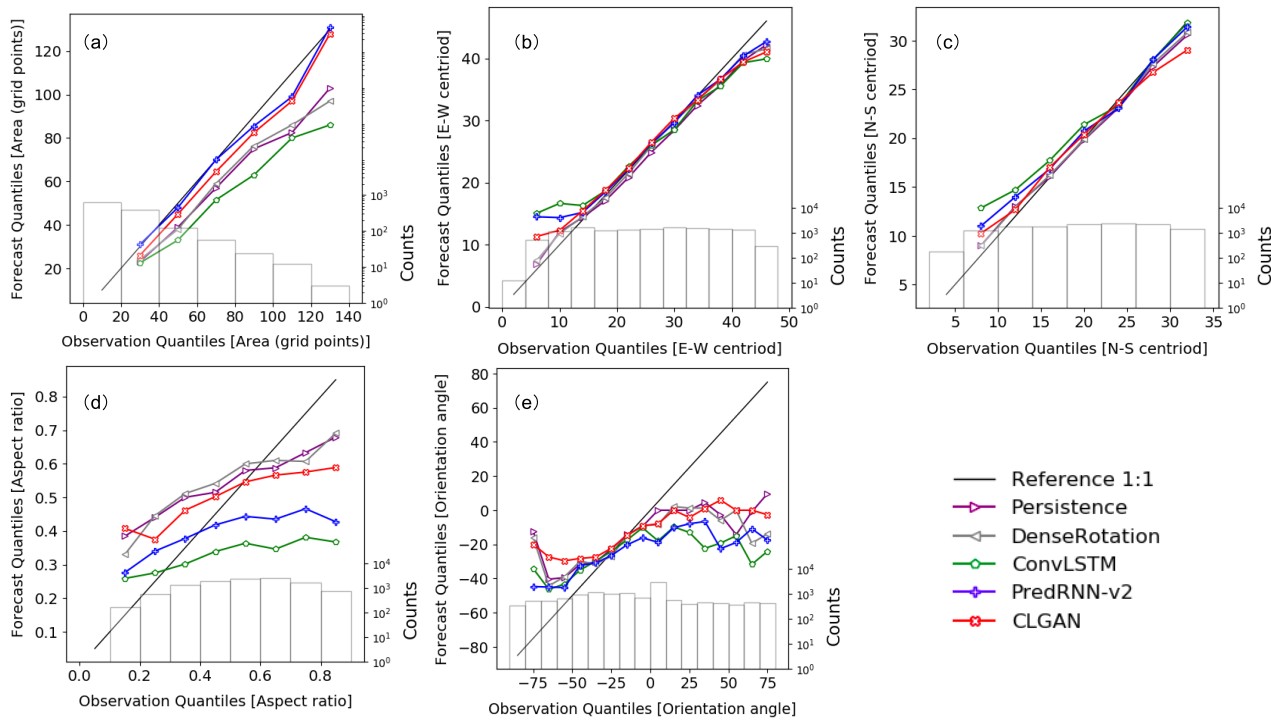

**Figure 5.** Conditional quantile plots in terms of the likelihood base-rate factorization for [(a) area, (b) E-W centroid and N-S centroid location, (d) orientation angle and (e) aspect ratio at a lead time of 60 minutes. The solid black line is the 1:1 reference line. The marginal distribution of the observations is presented as a histogram.

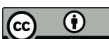



**Figure 6.** (a) A case study for a rain system moving from west to east while intensifying. The predictions of all models are illustrated with difference plots: (b) Persistence, (c) DenseRotation, (d) ConvLSTM, (e) PredRNN-v2 and (f) CLGAN. The initial time of the prediction period is 06:50 on June 12th 2019 (BJT).





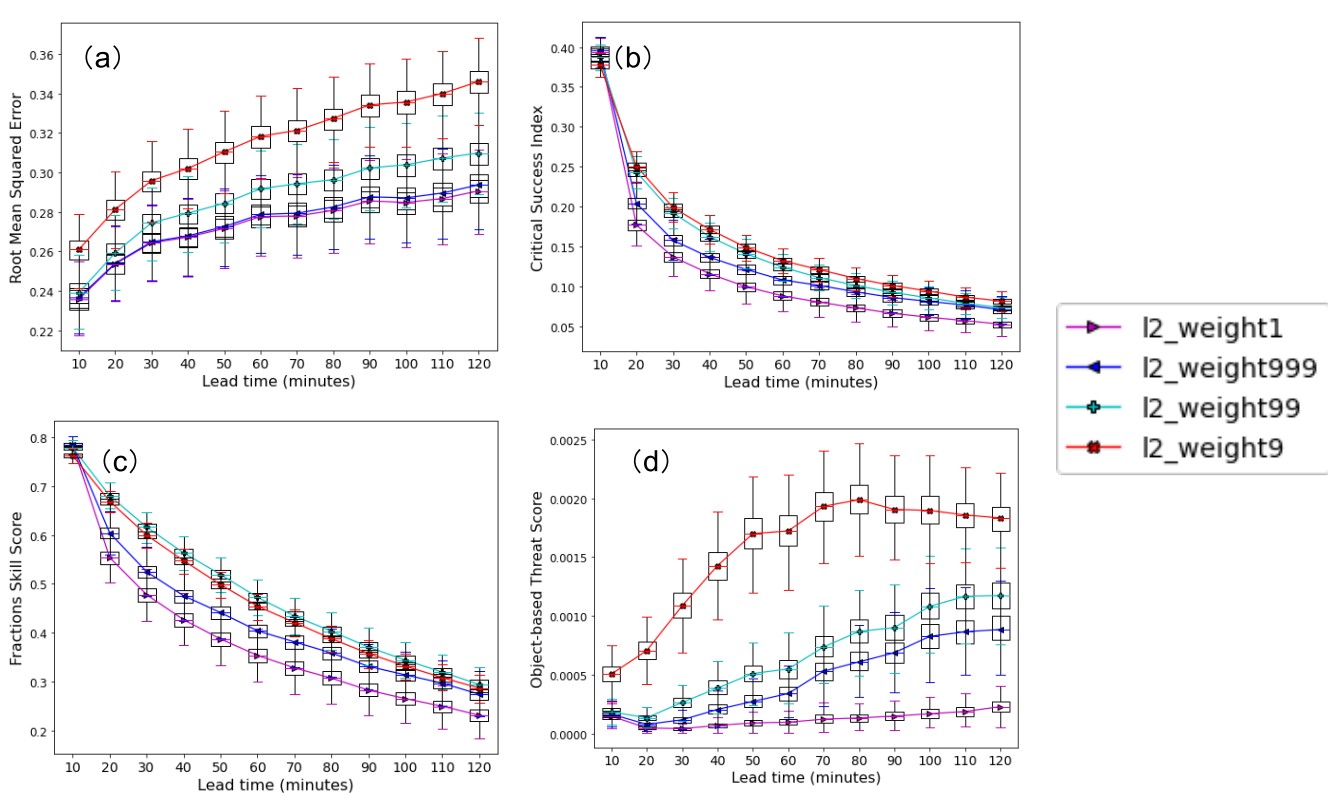

**Figure 7.** Mean scores [(a) RMSE, (b) CSI, (c) FSS and (d) OTS] for all lead times over the verification period for different weights of the $\mathcal{L}^2$ loss in CLGAN. The 'weight1', 'weight999', 'weight99' and 'weight9' on the legend mean the weights of the $\mathcal{L}^2$ loss are 1, 0.999, 0.99 and 0.9, respectively. As in Fig. 3, $t_{Pr} = 8\,\mathrm{mm/h}$ is chosen for CSI, OTS and FSS together with setting $s$ to 5 grid points for the latter. The area threshold $t_A$ of OTS is set to 9 grid points.



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
