# Peer review of "CLGAN: A GAN-based video prediction model for precipitation nowcasting"

_EGUsphere, 2022_

## Author Comment (AC2)

**General statement**

We would like to thank the editor for coordinating the review of our work and the peer-reviewers for their valuable comments on our study. In the following, we will address the referees' comments and present our plans and ideas for revising the manuscript. For clarity, our responses are highlighted in red.

**Referee comment #1**

Ji et al. present a method combining a generative adversarial network (GAN) with a U-Net and recurrent LSTM cells as the generator, for a high spatio-temporal resolution prediction of precipitation up to two hours ahead. The CLGAN model introduced in this work is compared against a set of baseline models and competing deep-learning based models with a comprehensive set of evaluation criteria. The work is well presented and the evaluation appears comprehensive, and is good for the scope of GMD. I only have minor comments before recommending this manuscript for publication.

**General comments:**

1. The authors use an "ablation study" (as defined in the abstract), or more specifically a sensitivity analysis on the weight of the GAN component, to assess the importance of the newly added GAN component for generating forecasts that pass the visual test and are more skillful in capturing the statistical properties of observed precipitation. This is a very useful contribution, although I feel that too many different technical terms are used in different places of the manuscript for describing this process (L8-10; L53-57; L73-74; L331-L342). I recommend the authors start with more general terms (e.g., "weight of the GAN-component") then use the terms reconstruction loss and adversarial loss after they are introduced in L189 eq. (3). Using it too early in the introduction may confuse readers.

Thanks for the comment and we will give more explanations of the used terms in the manuscript. We will start with the general terms "weight of the GAN-component" as suggested and then use specific terms "reconstruction loss and adversarial loss" after being introduced. Specifically, the reconstruction loss refers to the L1 or L2 loss and the adversarial loss refers to the score on the model to distinguish between real and generated data which is subject to mini-max optimization. The revision will be traceable in the manuscript.

2. Through looking at the code archived on Zenodo there appears to be code from a Git repository. Besides providing the Zenodo DOI could you also provide the GitLab repository link as well? This way potential users can follow the developments of the code and look at the README and documentation easier.

Thanks for the hint. We will add the gitLab repository link in the section of "Code and data availability".

**Specific comments:**

1. L73-74: As in general comment 1, try to avoid terms like "adversarial loss" before introducing

it. I also suggest being more specific about "sheds light on the interaction between the generator and the discriminator", e.g., the role of the GAN-component in generating forecasts with closer statistical properties of the observed precipitation.

Thanks for the comment and we will rephrase the sentence, i.e. A sensitivity analysis is performed on the contribution of the GAN-component in generating forecasts with closer statistical properties of the observed precipitation.

3. L161-162: The authors apply stronger weighting on higher precipitation rates to optimize towards heavy precipitation events. How much of an effect (if any) does this have on the precipitation of lighter precipitation events?

Thanks for the question. We calculated CSI with multiple thresholds of precipitation rate (0.1, 1, 2, and 8 [mm/hour]) and found that the proposed CLGAN model was competitive with the advanced model PredRNN-v2 for the nowcasting of lighter precipitation events, while both outperformed the Persistence, DenseRotation and ConvLSTM. PredRNN-v2 is slightly superior for shorter lead times (the first 60 minutes), whereas CLGAN outperforms for the longer lead times. Since we were more interested in the heavy precipitation forecasting, we only gave the evaluation of the precipitation events exceeding 8 mm/hour in the manuscript.

4. L372-374: It is mentioned that CLGAN can provide probabilistic forecasting by adding random noises. There is also mention of using ensemble forecasts to quantify forecast uncertainty in the introduction (L49-51). Is this part of future work or shown in this work? It was not very clear and I had to look if probabilistic forecasts were made in this work.

Thanks for pointing it out and sorry for the confusion. We didn't deploy a probabilistic forecasting framework in this study. This is mentioned in Line 372-374. However, it is correct that the proposed CLGAN can be used for probabilistic forecasting, by adding the noise as additional input. We acknowledge that a probabilistic nowcasting system is appealing due to the strong inherent uncertainties in the dynamics of precipitation patterns. This is especially true for small-scale convective precipitation systems which may produce extreme events. In this study, we focused on proving that a GAN-based approach is capable to circumvent the issue of forecasting too smooth precipitation patterns and investigate its sensitivity to the weighting terms in the loss function. The capacity of CLGAN to generate probabilistic forecasts will however be investigated in future studies.

4. Figure 1c. Is $L^G$ supposed to be $L^{GAN}$ here, to be consistent with text?

Thanks for pointing it out and we will revise it in Figure 1c.

5. Figure 6. For ease to read please label the subfigures (a)-(e) with subtitles (Observation, Persistence, DenseRotation, …)

Thanks for the comment and we will add the labels in Figure 6.

---

## Author Comment (AC3)

**General statement**

We would like to thank the editor for coordinating the review of our work and the peer-reviewers for their valuable comments on our study. In the following, we will address the referees' comments and present our plans and ideas for revising the manuscript. For clarity, our responses are highlighted in red.

**Referee comment #2**
The paper presents an interesting new method for nowcasting of precipitation, as well as a new precipitation dataset which could be useful for future machine learning applications. The work is thorough and well thought out, and would be suitable for publication in GMD after a few modifications and clarification listed below:

**Major Comments**
1. There are a couple of instances where the paper would benefit from an additional grammar check. This is especially noticeable in the abstract, the first few paragraphs of the introduction and the conclusion. I have highlighted some specific examples in my minor comments, but I would advise the authors to check the prepositions they use throughout the paper.

Thanks for the comment. We will carefully go through the whole paper and check the grammar issue. The revision will be traceable in the manuscript.

2. I find it odd that in Figure 3, the models cannot outperform the persistence forecast. Surely the ConvLSTM should be able to at least match the persistence as in effect one of its inputs is the persistence forecast. Do you have any intuition as to why you cannot beat persistence?

Thanks for the question. One possible reason why the models can barely beat the persistence forecast in the first 10 minutes is that the precipitation systems are relatively invariant within this very short time period. In our case, the Eulerian persistence forecast is the latest observations available, hence, which is highly correlated to the ground truth at short lead times. With lead times increasing, its performance degrade quickly. On the other hand, the models generate the forecasts by learning the temporal-spatial dependence from the sequential data, which could include fake textures or noise. It makes the models perform slightly worse than the Eulerian persistence at the first few time steps. But for the longer lead times, all the models are remarkably superior to the persistence.

3. In Figure 3c) and 3d) there is a very large degradation in skill at the 20 minute lead time for the ConvLSTM. As far as I can tell this is not discussed in the work and I think it needs to be discussed as it is quite a stark difference.

Thanks a lot for pointing it out. We will add the explanation in the Result section. One reason of the big difference is that the model performance is evaluated with the skill scores, which is affected by the choice of the reference model, rather than the original score values. Here, we choose the Eulerian persistence as our reference model and evaluate the CSI and ETS with a fairly

high threshold (8 mm/h). For the first time step (lead time of 10 minutes), both ConvLSTM and Eulerian persistence can capture strong precipitation events where ConvLSTM is even better. However, ConvLSTM models are prone to produce blurry predictions in an autoregressive prediction task, where the errors in the prior forecasts are inherited to the later ones. It shows that ConvLSTM becomes less efficient in capturing the strong precipitation events at the second time step while the Eulerian reference forecast still performs well due to the short lead time of 20 minutes. With increasing lead time, existing strong precipitation patterns in the Eulerian persistence get more and more displaced with respect to the ground truth data. Thus, the skill of the persistence forecast drops quickly (CSI and ETS are grid point-level metrics) and ConvLSTM again outperforms the persistence forecast, with positive skill scores.

4. Lines 286-288: It would be nice to see some discussion here as to why CLGAN is superior in terms of dichotomous forecast scores but not for RMSE. What attributes does it have or do other models not have, which help here?

Thanks for the comment. We will explain more about the results in this section. Indeed, the reason why the proposed CLGAN is superior in terms of dichotomous forecast scores, but not for RMSE is discussed in the ablation study (Line 339-342), where we found that the GAN-component encourages the model to generate forecasts which share similar distribution with the ground truth data, rather than just reducing the averaged pointwise loss. Hence, more heavy precipitation events are predicted by the CLGAN model which improves the dichotomous forecast scores. However, more predicted high-value precipitation could cause larger biases, compared to the models only generating low-value forecasts. The problem is magnified with the use of the point-by-point scores, i.e. RMSE, which suffer from the double penalty issue as we point out in Line 235-238.

5. Line 329: You say that CLGAN is doing better than ConvLSTM in the difference plot but it is different to see this in the eyeball norm. It would be useful to have some metrics even if it is just RMSE. Also please clarify what you mean by "fine cells"

Thanks for the comment. We will further evaluate the cases with quantitative metrics and add more details to the description in this section. The term "fine cells" here refers to small structures with high spatial resolutions. These small structures in the CLGAN forecasts indicate that CLGAN can generate more details of the precipitation system.

6. Line 331: The topic change here is very confusing as I thought you were still talking about the case study. Maybe add a sub-seciton title

Thanks for pointing it out. We will revise this part and make it more clear.

7. Lines 370-371 "It shows…": This sentence is very unclear. Have you already tried adding additional predictors? If so please provide a reference to this work. If this is future work then the c needs to be rephrased because currently it reads like this is a conclusion of the paper

Thanks for the comment and sorry for the confusion. The use of additional predictors is still

subject to the future work. We will rephrase it, i.e. "The idea is appealing since literature shows that the corresponding predictors and physical constraints can significantly improve the simulation of the targeted variable (Daw et al., 2017; Gong et al., 2022) and including additional predictors is subject to our future work."

8. Lines 371-372 "In addition…": Please explain further what you mean by expanding to probabilistic forecasting by adding random noise and how you could do this with your model.

Thanks for the comment. We will add more details about how to do the probabilistic forecasting with the proposed CLGAN model in this section. Indeed, GANs have been successfully developed for a probabilistic framework (Ravuri et al., 2021), by adding random noise (as perturbation) to the inputs which enables the generation of ensembles.

**Minor Comments**
1. Lines 6-8: This sentence is very difficult to understand and needs to be rephrased.

Thanks for pointing it out. We will revise the sentence as follows: "An efficient optical flow model, DenseRotation, as well as a shallow video prediction model, ConvLSTM, and an advanced one, PredRNN-v2, is performed as the competitors. A series of evaluation metrics, i.e. root mean square error, critical success index, fractions skill score and object-based diagnostic evaluation are used for a comprehensive comparison. We show that CLGAN outperforms the competitors in terms of scores for dichotomous events and object-based diagnostics."

2. Lines 73-74: The last highlight is very difficult to understand and needs to be rephrased

Thanks for the comment. We will revise it: "A sensitivity analysis is performed on the contribution of the GAN-component in generating forecasts with closer statistical properties of the observed precipitation."

3. Lines 100-102: This sentence needs to be rephrased

Thanks for the comment. we will rephrase it, i.e. "In our paper, the Eulerian persistence model and the DenseRotation model are used to show how well the traditional methods can perform for the precipitation nowcasting task and how much benefit can be further obtained by using DL-based video prediction methods."

4. Lines 120-121: This sentence needs to be rephrased

Thanks for the comment. We will revise the sentence, i.e. "One of them is the fully convolutional U-Net architecture, which is a U-shaped hierarchical encoder-decoder network with skip connections. The architecture enables abstraction of features on different spatial scales."

5. Section 3.1.4: It would be helpful to put references to figure 1 in this section because it is very

difficult to follow the CLGAN structure without references to the figure

Thanks for the suggestion. We will add the reference to Figure 1 here and make it easier for readers to follow.

6. Line 242: You are missing the index i in the expression and it is very confusing to have the forecast expression with the observation expression in brackets. You should separate out the two expressions.

Thanks for pointing it out and sorry for the confusion. We will revise the expression and make it more clear.

7. Figure 4: Please clarify in the text and figure caption what the box and whiskers represent

Thanks for the comment. We will add more details in the text and figure caption. Here, the box shows the range of the first quartile (upper) to the third quartile (bottom) of the scores, the whiskers are respectively the 95th percentile (upper) and 5th percentile (bottom) of the scores.

8. Figure 5: Add units to the legend and mention what the legend is in the caption

Thanks for pointing it out. We will add more details in the figure and caption. The metrics presented in Figure 5a-d are all unitless. Figure 5a shows the number of grid points of the observed and predicted precipitation objects, which has no units. Figure 5b-c show the distance between the centroid of the precipitation object and the western boundary (5b) and the southern boundary (5c). The distance is measured by the number of grid points, which still has no units. Figure 5d gives the aspect ratio of the precipitation objects (short side divide long side), which has no units. Figure 5e gives the orientation angle of the precipitation objects, which is the degree of the angle between the precipitation objects and positive x-axis. Positive values mean that the objects have a northeast-southwest orientation whereas negative values mean that the objects are oriented in southeast-northwest direction.

9. Figure 6: It is interesting that ConvLSTM does better at longer lead times here given that it does worse at longer lead times in the metrics. Do you have any intuition on this? Is it just a quirk of the case study you chose?

Thanks for the question. Actually, ConvLSTM is superior to the others in terms of RMSE and correlation coefficient (see in Figure 3a-b), even at the longer lead times. It indicates that ConvLSTM is capable of learning the precipitation patterns, as shown in the case plot (Figure 6). The drawback of ConvLSTM is that it is prone to generate blurry forecasts and not so efficient in capturing strong precipitation events, which is evaluated in terms of CSI and ETS (see in Figure 3c-d). Hence, it is reasonable that the ConvLSTM can perform fairly well in the case study but the proposed CLGAN is able to generate more detailed structures of the precipitation.

10. Line 344: This line needs to be rephrased

Thanks for the comment. We will remove the first sentence and rephrase it, i.e. "A novel architecture CLGAN is proposed in this work ..."

11. Line 358: More potential than what?

Thanks for pointing it out and sorry for the confusion. We will rephrase the sentence, i.e. "Compared to the conventional method, our results indicate that video prediction models with deep neural networks have a better capability to learn abstractions from data which in turn can improve the prediction of complex evolving systems."

12. Section 5: It would be interesting to have a comment about how you think your model would perform at longer lead times (say 6hrs). Would you still see such good results?

Thanks for the comment. We will add some comments on it in the Discussion section. Indeed, one reason why we focus on the nowcasting up to 2 hours is that the current NWP models have poor performance in this short range forecasting. With the lead time increasing, from 24 hours to 7 days, NWP still serve as the most efficient and reliable method for weather forecasting. Regarding your question, we expect that the model performance degradation will continue for longer lead times so that the skill will become comparable and later inferior to NWP models due to error accumulation in the auto-regressive prediction task. If we increase the lead steps from 10 minutes to 30 minutes, which means we still only have to predict the next 12 frames to obtain the 6-hour forecasts, there's a high chance that the superiority over NWP models can be prolonged. However, this comes at the price of a coarser temporal resolution.

**Bibliography**

Daw, A., Karpatne, A., Watkins, W. D., Read, J. S., and Kumar, V.: Physics-guided neural networks (pgnn): An application in lake temperature modeling, in: Knowledge-Guided Machine Learning, pp. 353–372, Chapman and Hall/CRC, https://doi.org/http://dx.doi.org/10.1201/9781003143376-15, 2017.

Gong, B., Langguth, M., Ji, Y., Mozaffari, A., Stadtler, S., Mache, K., and Schultz, M. G.: Temperature forecasting by deep learning methods, Geoscientific Model Development, https://doi.org/https://doi.org/10.5194/gmd-2021-430, 2022.

Ravuri, S., Lenc, K., Willson, M., Kangin, D., Lam, R., Mirowski, P., Fitzsimons, M., thanassiadou, M., Kashem, S., Madge, S., et al.: Skillful Precipitation Nowcasting using Deep Generative Models of Radar, arXiv preprint arXiv:2104.00954, https://doi.org/https://doi.org/10.1038/s41586-021-03854-z, 2021.

---

## Author Comment (AC4)

**General statement**

We would like to thank the editor for coordinating the review of our work and the peer-reviewers for their valuable comments on our study. In the following, we will address the referees' comments and present our plans and ideas for revising the manuscript. For clarity, our responses are highlighted in red.

**Community comments - Qiuming Kuang**

This paper presents a method of CLGAN(Convolutional Long short-term memory Generative Adversarial Network) for precipitation nowcasting. Experiment proves that the method is effective in capturing heavy rainfall events, which is very important for disaster prevention and mitigation. Meanwhile, the authors shared a precipitation data set from 2015 to 2019. This work is clearly presented. A few commends listed below:

1. DGMR (Skilful precision nowcasting using deep generating models of radar) is a SOTA algorithm for precipitation prediction using GAN method. DGMR uses radar echo data, while CLGAN does not use radar echo data. If conditions permit, it is suggested that CLGAN and DGMR methods can be compared. Otherwise, please compare and explain the advantages and disadvantages of the two methods.

Thanks for the comment. DGMR proves to be efficient and likely superior to the others. But in the meantime, the architecture of DGMR is highly complicated and sophisticated, which means that it is more difficult to understand the model performance since it consists of so many complex components. In our case, in addition to obtaining high-quality performance for the precipitation nowcasting task, we are committed to understanding the contribution of each component within the proposed deep neural network with the ablation study. Hence, we implemented CLGAN with the three comprehensible components: UNet, LSTM cells and GAN architecture. Literature and our study show that the hierarchical encoder-decoder network, UNet, is a powerful feature extractor on various spatial scales, the LSTM cells allow long-term information to be explicitly conveyed, and the GAN architecture encourages to generate predictions that share same statistical properties (distribution) as the ground truth. The ablation study further quantify the contribution of the GAN-component in simulating a highly uncertain system, i.e. precipitation.

2. In this paper, the authors point out that this method can improve heavy precipitation prediction. However, it is necessary to consider the strong radar echo, dynamic, water vapor, thermal and other environmental conditions in order to make a accurate heavy precipitation forecast. The authors are suggested to express this point.

Thanks for the comment. It's absolutely right that the additional predictors, i.e. the dynamic momentum, water vapor, thermal and other environmental conditions, can further improve the heavy precipitation forecast. One could obtain the environmental atmospheric state from the NWP models or extract the reflectivity data from the radar echos. But it is still a big challenge to merge all these information into the nowcasting task, especially when a rapid-updated forecasting system is required. A dense and real-time observation network, equipped with the regional NWP products

and remote sensing measurements, is useful for embedding the additional predictors in the DL-based models and further generating accurate heavy precipitation forecasts.

3. Figure 1 is somewhat miss-leading. In current version, the readers know how to get the t+1 th prediction using past m observations. However, the following n-1 frames are not provided. Certainly the results can be obtained iteratively. It is better to illustrate this explicitly.

Thanks for pointing it out and sorry for the confusion. We will replot Figure 1 to make it more clear and explicit.

4. In Figure 1, the input channel is c. It is not clear what is the actual number of c. And how many kinds of inputs are embedded.

Thanks for comment and sorry for the confusion. In this study, only the past $m$ precipitation observations are used as the inputs. Hence, $c$ equals to one in our study and we will mention it in the caption. We used the general number $c$ here because we are going to embed more additional predictors in our future work.

---

## Author Response (AR2)

**General statement**

We would like to thank the editor for coordinating the review of our work and the peer-reviewers for their valuable comments on our study. In the following, we will address the editor's and referees' comments and present our plans and ideas for revising the manuscript. For clarity, our responses are highlighted in red.

**Referee #2**

1. In response to Referee 2 comment 3, you give a lot of detail in your response which you have omitted in the paper. Please could you add this detail to the paper because I think it really helps with understanding the issue.

Thanks for the comment. We have added the explanation in the Result section (in Lines 287-294).

2. In response to Referee 2 comment 5, I don't think you have really answered my comment. You have added discussion about RMSE but I don't actually see any quantitative metrics, like, for example, the RMSE over all space.

Sorry for not presenting it in the previous revision. We gave the quantitative evaluation of RMSE in Lines 351-353 and CSI in Lines 355-366.

**Topical editor's suggestion**

1. Please perform a careful proofread of your manuscript as several grammar issues still remain.

Thanks for pointing it out. We will carefully go through the whole paper and check the grammar issue. The revision will be traceable in the manuscript.